# Structural Basis of Sequential and Concerted Cooperativity

**DOI:** 10.3390/biom12111651

**Published:** 2022-11-07

**Authors:** Veronica Morea, Francesco Angelucci, Jeremy R. H. Tame, Enrico Di Cera, Andrea Bellelli

**Affiliations:** 1Institute of Molecular Biology and Pathology (IBPM), National Research Council of Italy (CNR) c/o Dipartimento di Scienze Biochimiche “A. Rossi Fanelli”, Sapienza Università di Roma, Piazzale Aldo Moro 5, 00185 Roma, Italy; 2Department of Life, Health, and Environmental Sciences, University of L’Aquila, Piazzale Salvatore Tommasi 1, 67100 L’Aquila, Italy; 3Drug Design Laboratory, Graduate School of Medical Life Science, Yokohama City University, Tsurumi, Yokohama 230-0045, Japan; 4Edward A. Doisy Department of Biochemistry and Molecular Biology, Saint Louis University School of Medicine, St. Louis, MO 63104, USA; 5Dipartimento di Scienze Biochimiche “A. Rossi Fanelli”, Sapienza Università di Roma, Piazzale Aldo Moro 5, 00185 Roma, Italy

**Keywords:** allostery, heterotropic regulation, KNF model, MWC model

## Abstract

Allostery is a property of biological macromolecules featuring cooperative ligand binding and regulation of ligand affinity by effectors. The definition was introduced by Monod and Jacob in 1963, and formally developed as the “concerted model” by Monod, Wyman, and Changeux in 1965. Since its inception, this model of cooperativity was seen as distinct from and not reducible to the “sequential model” originally formulated by Pauling in 1935, which was developed further by Koshland, Nemethy, and Filmer in 1966. However, it is difficult to decide which model is more appropriate from equilibrium or kinetics measurements alone. In this paper, we examine several cooperative proteins whose functional behavior, whether sequential or concerted, is established, and offer a combined approach based on functional and structural analysis. We find that isologous, mostly helical interfaces are common in cooperative proteins regardless of their mechanism. On the other hand, the relative contribution of tertiary and quaternary structural changes, as well as the asymmetry in the liganded state, may help distinguish between the two mechanisms.

## 1. Introduction

Cooperative ligand or substrate binding and regulation of affinity by effectors are important functional properties of some enzymes, receptors, and transporters that allow these macromolecules to respond differently to different environments, a property that J. Wyman called the “cybernetics of biological macromolecules” [1], and J. Monod “the second secret of life”. Positive or negative homotropic cooperativity requires that the protein binds its ligand with stoichiometry higher than 1:1 and this is usually achieved because the protein is an oligomer made up of identical or similar subunits; the ligand affinity of the oligomer is then regulated by the number of bound ligands. Moreover, the ligand affinity of the oligomer may also be regulated by effectors binding to sites different from that of the principal ligand (heterotropic regulation). In 1963, Monod called “allosteric” the functional regulation achieved in enzymes by the binding of effectors to sites different from that of the substrate [2]; a later work Monod, Wyman, and Changeux gave a much more precise definition of allostery [3], which we adopt in this work (see below).

Homotropic cooperativity and heterotropic control can be achieved by several molecular mechanisms or types of mechanisms: sequential, concerted (as in Monod’s allosteric model [3]), or based on ligand-linked association/dissociation (these will not be considered in the present work). Sequential mechanisms hypothesize that ligation of one site causes the liganded subunit to change its tertiary structure (so called ligand-induced fit) and, consequently, its interactions with neighboring subunits [4,5]. Concerted mechanisms assume that the oligomer exists in at least two different, rapidly interconverting, quaternary structures having different ligand affinities, and that the ligands bias their equilibrium [3]. A graphical representation of the two reaction schemes for a homodimeric protein is reported in Figure 1. The mathematical expressions of the ligand partition of the two models are very different. Sequential models imply binding to one single state/structure of the macromolecule, with a progressive change in ligand affinity, e.g., for a tetrameric protein:*P*(*x*) = 1 + 4 *K**x* + 6 *α*
*K*^2^
*x*^2^ + *β*
*K*^3^
*x*^3^ + *γ*
*K*^4^
*x*^4^(1)
where *x* is the ligand activity, *P(x)* the sum of all ligation intermediates expressed as multiples of the unliganded one, K the equilibrium association constant for the (non-cooperative) binding step, and *α*, *β*, and *γ* are unit-less interaction factors that define positive (>1) or negative (<1) homotropic cooperativity at each step of ligation. Sequential models make structural predictions on the interaction factors; e.g., in the original Pauling’s “tetrahedral” model of Hb, which is the simplest possible model of this type, one has: *β* = *α*^3^, *γ* = *α*^6^.

The ligand partition function of the concerted model in Monod’s original formulation for a n-sites oligomer [3] is:*P*(*x*) = [(1 + *x**K**_R_*)*^n^* + *L*_0_ (1 + *x*
*K**_T_*)*^n^*]/(1 + *L*_0_)(2)
where we recognize two independent states (called *R* and *T*) that bind the ligand independently, non-cooperatively, and with different affinities (determined by the association constants *K_R_* and *K_T_*, respectively), whose equilibrium (in the absence of ligand) is governed by the allosteric constant *L*_0_ (notice that for consistency with Pauling [4] and Koshland [5], we converted the dissociation equilibrium constants originally preferred by Monod et al. [3] to association constants). Several variants of this reaction scheme are possible: e.g., in some cases the ligand affinity of the *T* state may be negligible, or the *T* state may present functional asymmetry and reduced binding stoichiometry, or the *T* state may be populated only in the presence of heterotropic effectors, in which case *L*_0_ is replaced by the function [Y]^T^ K_Y_, where ^T^ K_Y_ represents the equilibrium association constant of the effector Y to the unliganded *T* state protein.

The crucial differences between the two models are: (i) the concerted model imposes functional and structural symmetry (all subunits of the oligomer share the same structure/state), whereas the sequential model postulates asymmetry in the partially liganded derivative(s); and (ii) the concerted model postulates conformational equilibrium and population selection by the ligand, whereas the sequential model forbids this.

The main differences between the two models of cooperativity are summarized in Table 1. We remark that in Monod’s original paper [3], the term “allostery” is applied to the concerted reaction mechanism, considered alternative and incompatible with sequential binding, and that Koshland never used the term “allostery” to refer to the sequential reaction mechanism. Intermediate reaction schemes are impossible, but it is possible that the ligand binding reaction of one and the same protein sums up both mechanisms, e.g., because of an allosteric T-R equilibrium, where either state presents sequential cooperativity [6,7,8,9].

It is easy to recognize that readily accessible measurements are unable to determine whether a given cooperative protein obeys a concerted or a sequential mechanism; in particular, a structural difference between the fully liganded and fully unliganded state, in the absence of further analysis is not indicative. Moreover, it should be considered that both concerted and sequential phenomena may coexist in the same protein.

In this work, we explore two related hypotheses, namely (i) that a protein obeying the concerted mechanism may present a major quaternary structure change, coupled with smallish tertiary structure changes, whereas the opposite may occur in a protein obeying a sequential mechanism; and (ii) that a protein obeying a sequential mechanism, but not one obeying an allosteric mechanism, may present significant internal asymmetry in the partially, and possibly also in the fully liganded state. To explore these hypotheses, which represent the basic premises of the concerted and sequential models, it is necessary to assign the reaction mechanism independently of structural information; this can be done only in a handful of cases.

We analyzed the structural information available for a set of proteins whose reaction mechanism is known, albeit with different degrees of certainty, to identify possible structural features that may be common to proteins presenting concerted structural transitions and may differentiate them from proteins obeying a sequential reaction mechanism. The set of proteins we analyzed is summarized in Table 2; all of them present homotropic cooperativity, either of the positive or negative type; moreover, all of them are symmetric or quasi-symmetric at least in either the fully unliganded or fully liganded state. Since there are very few proteins presenting positive homotropic cooperativity for which the sequential mechanism is proven beyond doubt, we mainly selected as examples of this mechanism only proteins presenting negative homotropic cooperativity (see Table 1).

## 2. Materials and Methods

Structure analyses and computations were carried out using the atomic coordinates from the PDB [10] and the program Insight. For each protein, we tested several superpositions and measured the relative RMSDs of the *α*-carbons: (i) of the isolated subunits of the unliganded oligomer with each other to assess the intramolecular asymmetry of the unliganded state; (ii) of the isolated subunits of the fully liganded oligomer with each other to assess the intramolecular asymmetry of the liganded state; (iii) of the isolated subunits of the unliganded oligomer vs. the isolated subunits of the fully liganded oligomer, to assess the extent of the ligand-linked tertiary structure change; (iv) of the whole unliganded oligomer vs. the whole fully liganded oligomer, to assess the extent of the ligand-linked overall structure change. The RMSDs of the tertiary and quaternary structure changes are not additive, i.e., one cannot extract from these measurements the fraction of the RMSD due to the quaternary structure change; yet one can express the tertiary RMSDs relative to the overall RMSDs.

Where the structures of partially liganded derivatives were available, this was analyzed in the same way as those of the unliganded and fully liganded derivatives (see Table 3).

## 3. Results

### 3.1. Human Hemoglobin

Vertebrate Hbs, among which human Hb is the best studied example, are tetramers formed by two identical αβ heterodimers, called α1β1 and α2β2. Each subunit has a characteristic all-helical structure (the globin fold), made up of 8 helices named A through H [11]. The monomer–monomer interface of the heterodimer is isologous (i.e., symmetric [3]) and is contributed by helices B, G and H from each subunit, with contacts α1B-β1H, α1G-β1G, α1H-β1G. The interdimeric interface is again isologous, with contacts α1C-β2FG and α1FG-β2C, where FG indicates the interhelical segment interposed between helices F and G [12]. Several quaternary structures are known, only one of which has the low affinity characteristic of the *T* state [13]. The structural transition that converts the *T* into the *R* state can be described as a sliding and rotation of one dimer with respect to the other, with changes in the contacts at the α1β2 (and the symmetric α2β1) interface [6,12]. The isolated subunits and the αβ dimers have the same high affinity of the *R* state tetramer, thus Hb presents the quaternary constraint [14,15].

There is extremely convincing proof that human Hb behaves as a concerted system: (i) binding of heme ligands is cooperative, and heterotropic regulation is exerted by several effectors (protons, chloride, CO_2_, and 2,3 DPG); (ii) O_2_ binding in crystals or in silica gels, conditions that prevent the quaternary structure change, is non-cooperative and occurs with the low affinity characteristic of the *T* state [16,17]; (iii) the ligand-independent allosteric structure change has been recorded by a variety of means [18]; (iv) the structure of doubly liganded intermediates could be determined and is symmetric and *T* state (i.e., there is not any intermediate quaternary state nor any significantly asymmetric structure) [19,20,21,22].

Superposition of the subunits in the liganded and unliganded *T* and *R* state reveals that tertiary changes are very modest, and essentially limited to a small region of each subunit, composed by the F helix and the FG corner (the so-called allosteric core of each subunit) [23,24]; by contrast, quaternary structure changes are substantial (see Table 3 and Figure 2). Only approximately one fourth of the total RMSD of the α carbons between the liganded *R* structure and the unliganded *T* structure is accounted for by the tertiary structure change within each subunit, the remaining three fourths being accounted for by the quaternary, allosteric, structure change (see Table 4). Moreover, asymmetries in the structure of subunits within each quaternary state are very minor (Table 3). Indeed, the global (tertiary + quaternary) ligand-linked structure changes cause a RMSD (on the 𝛼-carbons) of 2.43 Å (*T*_0_ vs. *R*_4_), reduced to only 0.51 Å under conditions in which the quaternary structure change is prevented (*T*_0_ vs. *T*_4_) [25]. Ligand-linked tertiary structure changes, in the presence of quaternary transition, cause an average RMSD of 0.67 Å, corresponding to 28% of the overall structural change (Table 4). Ligand-linked tertiary structure changes, in the absence of the quaternary structure transition, cause an average RMSD = 0.28–0.41 Å in the *T* state and RMSD = 0.4 Å in the *R* state (measured for the β_4_ homotetramer, not reported in Table 3). Ligand-independent structure differences due to asymmetry are minor. All these observations are fully consistent with the expectation that in an allosteric, MWC-like macromolecule, the quaternary structure change dominates in amplitude over the tertiary structure changes, be they ligand-dependent or ligand-independent.

The case of Hb is also relevant for the important information made available by rapid kinetic experiments and molecular dynamic simulations, which consistently point out that quaternary (allosteric) structural changes occur over a time window at least 100-fold faster than tertiary structural relaxations. We comment on this point more extensively in Section 4.

### 3.2. Aspartate Transcarbamylase

Asp transcarbamylase (ATC) from *E. coli* presents sigmoidal activity curves, indicative of positive homotropic cooperativity for aspartate, heterotropic inhibition by CTP, and heterotropic activation by ATP; moreover, it presents large tertiary and quaternary structure changes originally detected because of a ligand-induced change in the sedimentation coefficient [26], and successively confirmed by X-ray crystallography. The presence of a ligand independent equilibrium between the *T* and *R* quaternary structure was demonstrated by fluorescence spectroscopy on a site-directed mutant enzyme having a small *L*_0_ [27]. Thus, ATC is a proven case of an allosteric enzyme obeying the concerted reaction mechanism.

The three-dimensional structure of the enzyme is peculiar and complex: it has a central core formed by two trimers of identical catalytic subunits, named C1 through C6. The intersubunit interfaces in each trimer are heterologous. Each trimer has rotational symmetry of 120° around an axis perpendicular to the plane of the subunits. The two trimers are arranged as a symmetric homodimer, by rotation of one trimer by 180° with respect to the other; and the trimer–trimer interface is isologous [28]. Due to this peculiar and highly symmetric arrangement, which we called a dimer or rings [29], a structural unit can be identified made up by two subunits, one from each trimer forming a symmetric homodimer; the interfaces between these homodimers are all isologous (we called them “extended isologous” to stress the fact that the unit required by the isologous interface is not a single subunit, but a oligomer, in this case a dimer). Three homodimers of regulatory subunits named R1 through R6 are located at the corners of the dimer of trimers. Surprisingly, the homodimer of regulatory subunits, in spite of having isologous interfaces, is asymmetric (see Table 3).

The quaternary structure change could be studied thanks to the competitive inhibitor PALA (N-phosphonacetyl-L-aspartate) which mimics Asp. In the *T* to *R* transition, the ring–ring interface undergoes a rotation of 12°; this is associated with a significant tertiary structure change in the catalytic subunits, whose two domains close on each other by 6.8° [28].

The structural mechanism of ATC cooperativity is more complex than that of Hb, as demonstrated by studies carried out on the substructure composed by the isolated trimer of catalytic subunits. The trimer has high ligand affinity, as characteristic of quaternary constraint, but the tertiary structure of the unliganded subunits in the trimer more closely resembles that of the *T* rather than the *R* state ATC, whereas the structure of the PALA liganded trimer resembles that of the PALA liganded *R* state ATC [30,31]. Thus, the ligand-linked tertiary structure change of ATC freely occurs in the non-cooperative trimer, and the role of the full dodecameric assembly would be that of modulating the flexibility of the catalytic trimers.

The essential results of our analysis of ATC are as follows: global (tertiary + quaternary) ligand-induced structure change causes a RMSD of 6.20 Å; tertiary changes in the presence of quaternary transition (catalytic and regulatory subunits) cause average RMSDs of 1.7–1.85 Å. The ratio between these two values (i.e., the fraction of the total RMSD that may be ascribed to ligand-dependent tertiary structure changes) is 0.28, identical, within errors, to the value obtained for Hb (see Table 3 and Table 4). Tertiary structure asymmetry of the catalytic subunits, in both the *T* and *R* states, is minor if compared to the overall ligand-dependent structure change. The regulatory subunits form a strongly asymmetric dimer in all states of the enzyme; thus, they should probably not be considered here. Thus ATC, like Hb, presents all the structural features that Monod and co-workers [3] postulated for a concerted, allosteric system.

### 3.3. Phosphofructokinase

Phosphofructokinase (PFK), the key regulatory enzyme of glycolysis, presents sigmoidal activity curves with respect to fructose-6P (but not with respect to ATP), and heterotropic regulation by ADP and P-enol pyruvate (PEP) [32]. Moreover, the enzyme from *B. stearothermophilus*, whose structure has been determined, presents a large quaternary structure change on substrate ligation that has been interpreted as a *T* to *R* transition; hence the enzyme is considered an example of an enzyme obeying the concerted reaction mechanism. Cooperativity in the binding of fructose-6P is only observed in the presence of inhibitors (PEP or its analogue P-glycolate, PGC). In the absence of substrates and inhibitors, the enzyme crystallizes in the putative *R* state; thus, PFK appears to be a case of inhibitor-induced allostery, and in equation 2 one should replace *L*_0_ with [PGC] ^T^ K_PGC_ (see Section 1). Unfortunately, no structure of any ligation intermediate is available.

PFK is a homotetramer whose four monomers (named A, B, C, and D) assemble in two dimers (AB and CD). All interfaces are isologous [33]. The putative T-R structural transition is described as a rigid body rotation of one dimer with respect to the other by approximately 7° [32]. A very interesting feature of the enzyme is that the binding sites of fructose-6P and PEP or its analogue PGC are located at the interfaces: those of F6P at the interdimeric interfaces, at the contact regions between monomers A–D (and the symmetry equivalents B–C); those of PEP/PGC at the intradimeric contact regions between monomers A–B (and the symmetry equivalents C–D). As is characteristic of isologous interfaces, each binding site occurs twice, thus the tetramer has four F6P and four PEP/PGC binding sites. The F6P and PEP/PGC binding sites are relatively close, and binding of each respective ligand alters both, thus explaining the inhibition by PEP; indeed, PFK presents the phenomenon of R-binding only, and apparently binding of PEP/PGC and F6P is mutually exclusive. An interesting structural feature of *T* state, PGC-inhibited, PFK is that, contrary to the PGC-free *R* state, it is asymmetric, both within the dimer and across the dimers (Table 3).

As typically found for multimeric proteins presenting ligand binding sites at the inter-monomeric (or inter-dimeric) interfaces, PFK presents quaternary enhancement for F6P, instead of quaternary constraint [34,35], and the movement of one dimer against the other caused by the quaternary structure change disrupts the F6P binding site. Quaternary enhancement is usually considered incompatible with the allosteric model [15,36]; however, it seems to us that the analysis by Edelstein and Edsall [37] may be extended to make quaternary enhancement compatible with the allosteric model. We give a simplified demonstration of this point in Figure 3 and its legend, which represents the relationships between ligation, conformational equilibrium and dissociation into dimers of an allosteric tetramer like PFK.

Our analysis of the ligand-induced structure changes gives the following results: global (tertiary + quaternary) changes cause a RMSD of 1.55 Å; tertiary structure changes, in the presence of quaternary transition cause an averaged RMSD of 0.94 Å. Ligand-independent tertiary structure differences due to asymmetry cause a RMSD of 0.28 Å in the *T* state, whereas the *R* state is symmetric (Table 3). It is important to remark that ligand-free PFK crystallizes in the *R* state, and the *T* state is induced by the binding of the inhibitor PEP or PGC. Thus, inhibitors convert the fully unliganded, symmetric *R* state tetramer into an asymmetric *T* state one. We suspect that in PFK, sequential and concerted phenomena may both be present, at different levels: sequential cooperativity either of the positive or negative type may affect the binding of PGC, as suggested by the *T* state asymmetry, whereas concerted cooperativity may be the major determinant of homotropic cooperativity for F6P in the tetramer. Consistent with the hypothesized coexistence of sequential and allosteric cooperativity, the extent of the ligand-induced quaternary changes is of the same order of magnitude as the tertiary ones instead of much larger as in the purely concerted systems of Hb and ATC.

A tetramer like PFK or Hb, which reversibly dissociated into dimers, presents the equilibria depicted in Figure 3. In this Figure, *R*_0_ indicates the unliganded *R* state tetrameric oligomer; *R*_4_ the fully liganded *R* state tetrameric oligomer; similar conventions are applied to the *T* state (*T*_0_ and *T*_4_) and to the dimer (*D*_0_ and *D*_2_). K_R_ represents the ligand association constant for the *R* state tetramer, *K_T_* for the *T* state tetramer, and *K_D_* for the dimer. *^T^ K*_D,0_ represent the equilibrium constant for the dissociation of the unliganded *T* state tetramer into dimers; similar conventions are applied to *^R^ K*_*D*,0_, *^T^ K*_*D*,4_, and *^R^ K*_*D*,4_. The allosteric constants for the unliganded and fully liganded states are labelled as *L*_0_ and *L*_4_, respectively.

The scheme, which for simplicity only includes the unliganded and fully liganded states, makes it obvious that:

(i) positive homotropic cooperativity demands that *K_T_* << *K_R_*, *L*_0_ >> 1, and *L*_4_ << 1.

(ii) *L*_4_ = *L*_0_
*K**_T_*^4^/*K**_R_*^4^; (iii) *L*_0_ = *^R^*
*K*_*D*,0_/*^T^*
*K*_*D*,0_ and *L*_4_ = *^R^*
*K*_*D*,4_/*^T^*
*K*_*D*,4_.

From these relationships, we obtain: *^R^*
*K*_*D*,0_ >> *^T^*
*K*_*D*,0_; *^R^*
*K*_*D*,4_ << *^T^*
*K*_*D*,4_; and *^R^*
*K*_*D*,4_/*^T^*
*K*_*D*,4_ = (*^R^*
*K*_*D*,0_/*^T^*
*K*_*D*,0_) (*K**_T_*^4^/*K_R_*^4^).

Quaternary enhancement implies that the isolated dimers/subunits have lower ligand affinity than the oligomer, whereas the opposite applies to quaternary constraint; given the above relationships, quaternary enhancement also implies that the liganded state is more tightly associated than the unliganded state (has lower tendency to dissociate into dimers), whereas the opposite occurs for quaternary constraint.

The allosteric model was originally conceived under the assumption of quaternary constraint (that at the time had been demonstrated only for human HbA). In the case of “perfect” quaternary constraint we have: *K_D_* = *K_R_*; and *^T^ K*_*D*,4_ = *^T^ K*_*D*,0_ (*K_R_*^4^/*K_T_*^4^). Thus, as pointed out by S.J. Edelstein and J.T. Edsall [37], the tetramer dimer dissociation constant of at least one of the two states can be ligand independent, whereas the other must vary with ligation. In perfect quaternary constraint, the tendency of the *T* state protein to dissociate increases with ligation until it exceeds that of the *R* state, causing L to decrease with ligation.

Although less straightforward, the opposite condition of “perfect” quaternary enhancement is not incompatible with allostery: indeed, in this case, we would have: *K_D_* = *K_T_* and *^R^ K*_*D*,4_ = *^R^ K*_*D*,0_ (*K_T_*^4^/*K_R_*^4^), and the tendency of the *R* state protein to dissociate decreases with ligation, causing the fully liganded *R* state to become more stable than the fully liganded *T* state, as it occurs in PFK.

For the above system, the law of mass conservation dictates:2 *D*_0_^2^/*K**_diss_* [(1 + *x*
*K**_R_*)^4^ + *L*_0_ (1 + *x*
*K**_T_*)^4^] + *D*_0_ (1 + *x*
*K**_D_*)^2^ − *D**_tot_* = 0(3)
where *x*, *K_R_*, and *K_T_* have their usual meaning, *K_D_* is the equilibrium association constant of the ligand with the dimer, *K_diss._* is the equilibrium dissociation constant of the tetramer into dimers in the absence of the ligand, *D_tot_* is the total protein concentration (activity) expressed on a dimer basis and *D*_0_ is the absolute molar concentration (activity) of the unliganded dimers. Solving this equation for *D*_0_ allows one to calculate the absolute molar concentration of each species in the reaction scheme, and to verify that, under appropriate experimental conditions, the model predicts allosteric homotropic cooperativity no matter whether *K_D_* = *K_R_* (pure quaternary constraint) or *K_D_* = *K_T_* (pure quaternary enhancement).

### 3.4. Glycogen Phosphorylase

Glycogen phosphorylase is a very interesting example of a putative concerted enzyme. The non-phosphorylated enzyme (GPb) is activated by phosphorylation (GPa). The quaternary structures of its functionally most important states are the homodimer and the less active homotetramer. Glycogen or oligosaccharides promote dissociation of tetramers to more active dimers. The putative T-state and R-state structures have been characterized; the former is a symmetric homodimer, the latter a homotetramer, described as a dimer of homodimers, whose monomers present minor asymmetries. Thus, the structural comparison is usually carried out between the *T* and *R* state homodimers. The 850-residue subunit of mammalian GP has an α/β structure; it contains a PLP cofactor, required for catalysis. The structure includes two domains, the larger N-terminal being responsible for the intersubunit interface. The intersubunit interface is isologous and is provided by two contact regions formed by α-helices 7 (the “tower” helices; residues 262–276 in rabbit muscle GP; see Figure 4) of each subunit, which contact each other in antiparallel directions, and the cap region of each subunit (residues 35–46) contacting the α-helix 2 (residues 47–78) of the opposite subunit [38,39,40].

The quaternary structure change may be described as a rotation of one monomer with respect to the other by 10°; this movement causes the cap’-α-helix2 contact region to become tighter by 1 Å and that between the tower helices to move apart by 3 Å [38]. Moreover, in the *T* quaternary structure the two tower helices are oriented in an almost (anti)parallel direction, forming an angle of −20°, whereas in the *R* quaternary structure they are almost perpendicular to each other (angle = −80°). The catalytic site is quite far from the interface, but the C-terminus of the long tower helix is contiguous with residues that gate the access to the catalytic site (Pro 281 and Asp 283).

The binding sites of some heterotropic effectors (Ser 14 which can be phosphorylated; inorganic phosphate), lie in close proximity of the cap’-α-helix2 contact region and this explains their effect on the quaternary structure change.

Unfortunately, there are no available structures of ligation intermediates of GPb, and the ligand-independent allosteric interconversion of the two quaternary structures has not been demonstrated, thus it cannot be unequivocally confirmed that GPb obeys a pure concerted reaction mechanism. Moreover, as demonstrated by the structural parameters reported in Table 3 and Table 4, GPb presents *R* state asymmetry, a feature that is unexpected in an allosteric protein obeying the concerted mechanism, and characteristic instead of proteins obeying a sequential reaction mechanism. The global (tertiary+quaternary) ligand-induced structure change has RMSD = 2.61 Å; tertiary structure change, in the presence of quaternary transition has an average RMSD = 1.33 Å; the ratio of the latter to the former yields ≃ 50%, a value higher than that we measured for pure allosteric macromolecules like Hb or ATCase, but similar to that of PFK (see Table 4). This finding, and the liganded state asymmetry, may suggest that the homotropic cooperativity of GP could result from the addition of a sequential component to the concerted mechanism, possibly to be envisaged as sequential cooperativity within the *R* state.

### 3.5. The Chemotactic Asp Receptor from S. typhimurium and E. coli

In many bacteria, the chemotactic aspartate receptor is a homodimeric protein exposed on the cell surface that presents negative homotropic cooperativity for Asp. Koshland and co-workers described the cases of the Asp receptor from *E. coli* and *S. typhimurium* [41,42,43]; both are homodimers presenting negative cooperativity, which in the case of the *E. coli* receptor is so marked to induce half-of-the-sites reactivity. The extracellular domain of *S. typhimurium* Asp receptor is of special interest since the unliganded and (singly) liganded structures have been solved (PDB entries 1LIH and 2LIG). The two monomers of the homodimer have a four-helix bundle structure, and the dimer presents isologous contacts mainly between helices 1 and 4 from each monomer. In the absence of Asp, the homodimer forms a perfectly symmetric structure, but the binding of Asp causes the homodimer to become asymmetric, and the affinity of the two binding sites, which reside in close proximity of the monomer–monomer interface, diverges by 20-fold [42]; moreover, the soluble domain of Asp receptor presents quaternary enhancement, its tendency to dissociate into monomers being diminished in the presence of saturating Asp [44]. In the presence of Asp, the extracellular domain of the *S. typhimurium* receptor presents complete occupancy of the high affinity site and partial occupancy of the low affinity site. Asymmetry is present both in the tertiary and quaternary structure of the liganded macromolecule: indeed, the two monomers of the liganded homodimer are no longer rotated by 180° with respect to each other but only by 178°. The ligand-induced quaternary structure change causes the two monomers to become more parallel and closer to each other, more markedly at the extremity that is farther from the membrane crossing domain. Interestingly, in this case, the interfaces are isologous but non-symmetric, because the interatomic distances differ even though the interacting residues are the same. This effect is amplified if one measures inter-residues distances for residues that are farther from the monomer–monomer interface; e.g., the distance between CαAla85 A-CαThr154 B is 23.2 Å in the unliganded state and 19.9 Å in the liganded state; the values for the distance between CαAla85 B-CαThr154 A are 23.2 Å and 13.9 Å, respectively (A being the high affinity subunit, B the low affinity one).

The Asp binding site lies across the monomer–monomer interface and both subunits contribute to it [43,45]; the relevant residues are Arg64, Ser68, and the segment 149–154 from one subunit and Arg69 and Arg73 from the partner subunit. Ligand binding to one site causes the other site to shrink, making a poorer fit for the second Asp ligand (see Figure 5).

Our analysis yields the following results: global (tertiary+quaternary) ligand-induced structure change has RMSD = 2.85 Å. The unliganded state of the homodimer is symmetric, whereas the liganded state is asymmetric, and one of the two subunits presents incomplete ligand saturation, nearing half-of-the-sites reactivity. Ligation of the first molecule of Asp causes a tertiary structure change with RMSD in the order of 2.5 Å in both subunits (Table 3); the structure difference of the two subunits in the half-liganded state is assigned a RMSD = 2.44 Å. Consistent with the expectations for a protein presenting sequential (negative) homotropic cooperativity, the quaternary structure change in the Asp receptor, though evident, is responsible for a small fraction of the total RMSD between the liganded and unliganded structures, the largest structure changes occurring at the tertiary level; moreover, the asymmetry (only present in the liganded state) causes a large tertiary structure difference between the two subunits within the structure of the half-liganded state. These features may be considered typical of the sequential reaction mechanism.

### 3.6. Asp Semialdehyde Dehydrogenase

Asp semialdehyde dehydrogenase (ASADH) is a homodimeric bacterial enzyme presenting negative cooperativity. The structures of ASADH from *E. coli* in the substrate bound and free state are available, with pdb codes 1T4B or 1T4D, 1GL3 [46]. The enzyme presents several very interesting features. The monomers have a predominantly β-sheet structure, and the isologous monomer–monomer interface is provided by two extended β-sheets; this is at variance with the other proteins considered in this work, which have predominantly helical interfaces. Each monomer is composed of two domains, each of them further split into two sub-domains (labelled N1, N2 and C1, C2). Each monomer presents one binding site for NADPH and one for ASA; however, while two molecules of the former can simultaneously bind to the dimer, only one of ASA can bind, due to the half-of-the-sites reactivity of the enzyme. The authors suggest that this property is important for the release of NADP^+^, and that asymmetry occurs in an alternate fashion, one site being charged with the substrates, while products are released from the other site.

The substrate (s) free enzyme is symmetric (closed form; pdb code 1GL3); the enzyme containing two molecules of NADPH and one of the ASA analogue S-methyl-L-Cys sulfoxide (SMCS) is asymmetric [46]. NADPH binding causes a set of symmetric concerted tertiary structure changes that can be described as reciprocal rotations and torsions of the sub-domains and domains of each subunit. SMCS binding to one subunit causes the movement of a loop towards the substrate which forbids the same movement in the opposite subunit (open form; pdb codes 1T4B and/or 1T4D).

Our analysis yields RMSD = 1.49 Å for the global (tertiary + quaternary) ligand-induced structure change, to be compared with RMSD = 0.94–1.3 Å for the tertiary structure change. Ligand-independent tertiary structure differences due to asymmetry are described by RMSD = 0.27 Å, in the unliganded state, and RMSD = 0.62 Å in the half-liganded state (structure 1T4B presents SMCS on subunit A only because of half-of-the-sites-reactivity). These values are reminiscent of those of the Asp receptor and should be interpreted along similar lines.

### 3.7. Bacterial D-Lactate Dehydrogenases

Bacterial lactate dehydrogenases (LDHs) are extremely interesting enzymes that would deserve a dedicated analysis. They belong to two distinct groups, with respect to their product: L-lactate or D-lactate. Bacterial L-lactate dehydrogenases are usually cooperative homotetramers that seem to obey an allosteric, MWC-like reaction mechanism. D-lactate dehydrogenases may be homodimeric or homotetrameric, may or may not present homotropic cooperativity and in the best characterized examples, they obey a sequential reaction mechanism.

Here we analyze only two examples of D-lactate dehydrogenases from Gram negative bacteria, whose three-dimensional structure is available: those from *Pseudomonas aeruginosa* (PaLDH) and from *Fusobacterium nucleatum* (FnLDH) [47]. Both are homotetrameric enzymes that present homotropic cooperativity, positive in the case of FnLDH (Hill coefficient for pyruvate, at constant NADH and pH = 8: *n* = 2), negative in the case of PaLDH (Hill coefficient for pyruvate, at constant NADH and pH = 8: *n* = 0.77). FnLDH has a special interest in this analysis because it is the only protein considered in this work that presents positive homotropic cooperativity and can be confidently assigned a sequential reaction mechanism [47].

Our analysis yields the following results: global (tertiary+quaternary) ligand-induced structure change has RMSDs = 6.44 Å for FnLDH and 3.30 Å for PaLDH; tertiary structure change has RMSDs = 4.7 Å for FnLDH and 2.35 Å for PaLDH. Both enzymes present significant ligand-independent tertiary structure differences due to asymmetry, but the description is this phenomenon is complex, and different in the two cases. Ligand-free (apo) PaLDH (PDB code 6ABJ) is a symmetric dimer of slightly asymmetric homodimers; the two subunits of the same dimer present RMSD = 0.31 Å. Ligand-bound PaLDH (PDB code 5Z20; the molecule contains NADH and oxamate, an inert analog of pyruvate) is an asymmetric tetramer in which no two subunits are identical, with an average RMSD = 0.5 Å. Ligand-free FnLDH (PDB code 6ABI) is a symmetric dimer of strongly asymmetric homodimers; the two subunits of the same dimer present RMSD = 1.9 Å. Contrary to PaLDH, the asymmetry is significantly diminished by ligation of substrates, and the four subunits of the liganded homotetramer (PDB code 5Z21) present an average RMSD = 0.25 Å. Thus, it seems that, contrary to the expectations, asymmetry in FnLDH is induced by de-ligation, rather than by ligation.

### 3.8. Bacterial Purine Nucleoside Phosphorylase

Purine nucleoside phosphorylase (PNPase) is an important enzyme participating in the purine salvage pathway. The enzyme activity is ubiquitous, but the bacterial and eukaryotic variants are unrelated. Bacterial (*E. coli*) PNPases are homohexamers, assembled as trimers of homodimers [48,49,50], whose structure corresponds to the one we previously characterized as rings of dimers [29]; all monomer–monomer interfaces are isologous, and two types of interfaces exist: one intra-dimeric, the other inter-dimeric. The “functional” dimer is clearly identified by the fact that the substrate binding sites are located at the intra-dimeric interface. The enzyme binds two substrates, namely phosphate and the nucleoside; the binding sites are adjacent to each other. The phosphate binding site provided by residues Arg24, Arg87, Ser90 and Gly20 plus Arg43 from the neighbouring subunit. The ribose binding site involves Ser90, Met180, and Glu181, plus His4 from the neighboring subunit [49]. The purine base is exposed to the solvent.

The structure of the subunit is composed by a central 8-stranded β-sheet core surrounded by 8 α-helices; the last helix (helix 8) can assume three conformations: it can be continuous, opening access to the substrates binding site, or present an interruption and a bend, closing over the substrates binding site; in some derivatives, it can be partly destructured (Figure 6).

The unliganded enzyme (PDB code 1ECP) [48] presents a mild intramolecular asymmetry, and pairwise superposition of the six subunits yields RMSDs in the order of 0.32 Å. In all subunits, the C-terminal helix (helix 8) is continuous, and the substrates’ binding site is open. Upon binding of substrates, a very interesting structure change occurs, both at the tertiary and quaternary level. The homohexamer acquires a binary symmetry axis and is formed by two symmetric units containing one dimer and one half-dimer each (PDB code 1A69) [49]. The three subunits in the asymmetric unit differ largely in tertiary structure: one (subunit A) presents an interruption in helix 8, closing the access to the substrate binding site; the other two (B and C), which form the functional dimer are both in the open conformation, in spite of the presence of the substrates in all three subunits. Accordingly, the three-subunit assembly is strongly asymmetric, the subunit presenting the split helix 8 differing from the other two by RMSD = 1.32 and 1.76 Å; and the two subunits presenting the continuous helix 8 differing from each other by RMSD = 0.98 Å (Table 3). The tertiary structure differences among the subunits in the liganded PNPase are greater than those between the liganded and unliganded PNPase (Table 3), and actually, the RMSDs are better explained by the open or closed structure of the subunit than by the presence or absence of ligands: open (liganded or unliganded) vs. closed (always liganded) has approximated RMSD = 1.0–1.3 Å.

More recently, a structure of partially substrate liganded PNPase was obtained, in which the hexamer contains 6 molecules of phosphate (or sulfate) and only 2 molecules of the nucleoside analogue Formycin A (PDB code 4TTA) [51]. The structure of this derivative is intermediate between those of the fully unliganded and fully liganded PNPase, because: (i) the homohexamer does not present symmetry axes, as it occurs in the fully unliganded PNPase; (ii) the functional dimers are AD, BE, and CF; interdimeric contacts occur between subunits A–F, B–D, and C-E; two subunits (A and F) present the closed conformation, two (B and D) the open conformation, as it occurs in the fully liganded PNPase, and the last two (C and E) present a partially broken/destructured helix 8; thus, the inter-dimeric contacts impose greater symmetry than the intra-dimeric ones; (iii) the destructured part of helix 8 is not resolved in subunit C, thus this subunit was not included in the analysis of RMSDs; and (iii) the two molecules of the purine nucleoside analogue are bound to the closed subunits, confirming the expectation that the closed structure has higher affinity for the substrate. The presence of open and closed subunits in the fully and partially liganded states of the enzyme is consistent with the observed negative cooperativity. The destructured helix 8 causes a tertiary conformation that differs strikingly from both the open and closed ones with RMSDs of 2.8 and 3.5 Å, respectively (Table 3).

The distribution of the open and closed (and destructured) subunits in fully and partially liganded *E. coli* PNPase obeys a remarkable rule: closed subunits belong to different dimers, but are adjacent, i.e., they contact each other via the inter-dimeric interface. This distribution is achieved because of symmetry or quasi-symmetry of the subunits connected by inter-dimeric interfaces, in spite of the strong asymmetry between subunits connected by the intra-dimeric interface (i.e., belonging to the same dimer). This distribution suggests a hitherto unrecognized functional relevance of the hexameric assembly. Moreover, the ordered distribution of the asymmetric subunits in liganded PNPase explains the otherwise perplexing fact that the average RMSD for the superposition of subunits is larger than the overall RMSD for the superposition of the whole assemblies: indeed, some subunits never superimpose when superpositions are explored for the whole hexamers.

### 3.9. Other Enzymes Obeying a Sequential Reaction Mechanism

dTMP synthase is the homodimeric, negatively cooperative enzyme that catalyzes the final step of dTMP biosynthesis; its substrates are dUMP and methyl-tetrahydrofolate. The enzyme presents strong negative homotropic cooperativity, with half-of-the-sites reactivity: in the presence of excess dUMP and the THF analogue CB3717, the homodimer binds two molecules of dUMP but only one of CB3717. The binding site(s) of substrates are located at the monomer–monomer interface [52]. The (half-) liganded enzyme is asymmetric, the two monomers differing with RMS = 0.43 Å. Unfortunately, to our knowledge, the structure of the unliganded enzyme is not available, thus we could not carry out a complete analysis.

Glyceraldehyde-3-phosphate dehydrogenase (G3PDH) is a homotetramer presenting negative cooperativity for NAD^+^, which binds to only two subunits, confirming the half-of-the-sites reactivity of the enzyme [53]. Indeed, G3PDH is noteworthy because, at least in the enzyme from yeast, the first two molecules of NAD might bind with positive cooperativity, whereas the last two exhibit negative cooperativity [54].The subunits of the tetramer bearing 2 molecules of NAD^+^ (PDB code 1J0X) present mild asymmetry, with an average RMSD of 0.31 Å. Quite surprisingly, the asymmetry is extended to all the four subunits, i.e., the tetramer cannot be described as made up of two identical but asymmetric dimers, each bearing one molecule of NAD^+^. Here again, we did not find any structure for the fully unliganded enzyme.

Vivoli et al. [55] make an interesting case for bacterial heptose isomerase, a tetrameric enzyme that may present both positive and negative homotropic cooperativity, in the absence of a clear-cut quaternary structure change; however, we feel that their study, though promising, does not include enough information to be included in the present analysis.

## 4. Conclusions

The aim of this work is to compare examples of the structures of cooperative proteins obeying either the MWC or the KNF reaction mechanism and to look for structural features that are common to either group or which may discriminate between the two groups. We resorted to a model-free analysis based on the RMSDs, because more refined analyses based on the measurement of “movements” of subunits or domains cannot be easily generalized to the case of structurally unrelated proteins. Our work has several limitations essentially because of the limited number of proteins for which sufficient information is available. Indeed, we required: (i) a reasonable certainty about the ligand binding reaction, whether sequential or concerted; and (ii) the 3D structures at atomic resolution of at least the unliganded and fully liganded states; possibly also of partially liganded intermediates. The first requirement is particularly demanding because sequential and concerted reaction mechanisms are both compatible with positive cooperativity and distinguishing among them is based on assumptions that, despite being very clearly defined, may be difficult to demonstrate empirically. The essential features that distinguish the MWC from the KNF model are listed in Table 1. Assessing these features is less straightforward than it may appear at first sight because the KNF reaction scheme requires that the incompletely saturated reaction intermediates break the structural symmetry of the oligomer, but makes no assumption on the fully unliganded and fully liganded derivatives. In the majority of cases, only the structures of the fully unliganded and fully liganded protein are available, while those of ligation intermediates, which would be more informative, are usually not available, except for proteins presenting negative homotropic cooperativity. Two distinguishing criteria are: (i) the demonstration of a ligand-independent structure change, promoted by heterotropic effectors or other experimental conditions strongly suggests the concerted mechanism; (ii) negative homotropic cooperativity, is consistent with the concerted mechanism only if subunit inequivalence or a specific (and uncommon) breakage of symmetry are present in the *R* state, thus as a general rule is an indicator that some variant of the KNF reaction scheme is obeyed. Quaternary structure differences between the fully liganded and fully unliganded states, per se, do not imply either the MWC or KNF reaction scheme, but are compatible with both. Moreover, the possibility should be considered that the same protein might present both types of phenomena; we consider this possibility as the sum of the two mechanisms, rather than some intermediate between them.

This study represents a first attempt to quantify some aspects of structure-function relationships of cooperative proteins at a group level, rather than going into detail on single cases; in particular, since only a few examples were available for each group, we cannot pretend to generalize our results; moreover, we excluded possible examples of KNF-like proteins present positive homotropic cooperativity (with the exception of FnLDH), because of the criteria given above. Nonetheless, our analysis offers some general suggestions:

(i) All the proteins here considered, except PNPase, are homodimers, either homodimers of monomeric subunits or homodimers of more complex assemblies (e.g., of heterodimers, as in the case of Hb or of trimers, as in the case of transcarbamylase). All of them, PNPase included, present isologous interfaces [3].

(ii) In the absence of ligands, all the proteins considered in this work have symmetric or quasi-symmetric quaternary structure, with the exception of FnLDH, whose unliganded state presents significant asymmetry (see Table 3). In their liganded state they are symmetric or quasi-symmetric if they obey a concerted reaction mechanism, asymmetric if they obey a sequential one. Glycogen phosphorylase b is an outlier in this respect because it is a reputed allosteric protein but presents *R* state asymmetry.

(iii) Of particular interest are the partially liganded derivatives whose structures are available in a minority of cases. In Hb, the doubly-liganded derivatives of known structure are symmetric and have the same quaternary structure of the unliganded protein; however, one should consider that all such derivatives whose structure is available, have one ligand per heterodimer, a factor that may favor symmetric structure. In all other cases in which the structure of partially liganded derivatives is available, the protein presents negative cooperativity, thus, it is not compatible with the MWC model. In these cases, the partially liganded protein is asymmetric, consistent with the expectation of the KNF model. Interestingly the fully liganded derivative may often remain asymmetric.

(iv) The tertiary structure changes responsible for cooperativity are small both in MWC-like and KNF-like proteins, and limited to small portions of the subunit. Quaternary structure changes favored by ligation are usually large in proteins obeying the concerted reaction mechanism, as predicted by the MWC model, and usually small in sequential-binding proteins obeying a KNF-like reaction scheme (Table 4). We can rationalize this finding by saying that in a macromolecule that maintains symmetry, a large quaternary structure change is instrumental to obtain the tertiary rearrangements responsible for the changes in ligand affinity; by contrast, symmetry violations may be achieved by limited, if any, quaternary structure rearrangements and local tertiary conformational changes. For example, in the case of Hb, a 1 Å change in the distance between the C helix and FG corner within each subunit [56] is obtained by having the βFG corner sliding over the αC helix by one helical turn, a movement of approximately 6 Å [12]. If we compare the tertiary and overall ligand-dependent structure changes (Table 4), we observe that, in our set, in the proteins obeying a sequential reaction mechanism, tertiary changes correspond to ≥70% of the overall RMSD, whereas in the proteins obeying a demonstrated concerted mechanism, tertiary changes account for only ≤30% of the overall RMSD, the remainder being accounted for by the quaternary structure changes. These estimates should be taken as indicative, given that the tertiary and quaternary contributions to RMSDs are not additive. The cases of PFK and GPb do not fit the above estimates, but probably in these enzymes concerted and sequential events coexist.

(v) α-helices play a very important role in the ligand-linked structure changes we observed (see Figure 4 and Figure 5).

(vi) Isologous interfaces are compatible with asymmetry. This occurs because monomers forming a perfectly isologous interface in the absence of the ligand may undergo subtle tertiary structure changes that prevent some of the interface contacts to remain perfectly symmetric: e.g., a loop may move and prevent the equivalent loop on the symmetry-related subunit from moving in the same way; or an intersubunit contact may be tighter in one place, and looser at the symmetry-equivalent site. These effects may be exquisitely subtle.

(vii) In the structures we analyzed above, it is quite common that the ligand binding site occurs at a monomer–monomer interface, and interacts with residues from both monomers; this is the case for several proteins in our analysis, e.g., PFK (reputedly a MWC protein), and the Asp receptor. Ligand binding at the interface is associated with quaternary enhancement and is not incompatible with the allosteric MWC model.

(viii) In the half-liganded state of proteins that present negative homotropic cooperativity, we observed not only an asymmetry of the constituent monomers, but also tertiary structure differences between both monomers and the monomers of the unliganded state (see Table 3), i.e., ligand binding to one subunit changes the structure of both subunits albeit in different directions. In some cases, these changes are exquisitely ordered within the oligomer (e.g., in PNPase).

(ix) In our set of enzymes, allosteric or reputedly allosteric ones present heterotropic regulation in addition to homotropic cooperativity; heterotropic regulation is usually absent in the enzymes obeying a sequential reaction mechanism (except by hydrogen ions). There are however some examples of enzymes presenting negative homotropic cooperativity and heterotropic regulation (e.g., glyceraldehyde 3-phosphate dehydrogenase and deprenyl derivatives [53]), which would deserve a future study. This observation establishes a link between Monod’s two definitions of cooperativity [2,3].

(x) Allosteric enzymes, as predicted by Monod et al. [3], usually catalyze one among the initial reactions of a metabolic path. The case for enzymes obeying a sequential mechanism is not straightforward due to the fact that we selected them because of negative homotropic cooperativity, which, as suggested by Cornish-Bowden [57], is confined to the central or terminal reactions of metabolic pathways.

Our “static” structural analysis may be compared with some pertinent information coming from rapid kinetic experiments and molecular dynamics simulations. In the case of Hb, both rapid photolysis experiment and molecular dynamic simulation consistently demonstrate that the allosteric quaternary structure change occurs over a time window of tens to hundreds of microseconds [58,59], whereas the first tertiary relaxations that follow photochemical de-ligation occur over a time window of hundreds of nanoseconds. This is confirmed by molecular dynamic simulations [60]. Few comparative analyses on different proteins have been carried out, e.g., [61]. It is difficult to generalize the results obtained thus far, but it is tempting to propose the hypothesis that the rate of the structural changes following de-ligation might be indicative of the reaction mechanism, the expectation being that proteins obeying a sequential mechanism (which require mainly tertiary structure changes) would relax much faster to their equilibrium unliganded conformation than proteins obeying a concerted mechanism (which requires substantial quaternary structure changes).

It is interesting to compare the results presented here with some previously published studies. Daily and Gray [62] developed a method to analyze tertiary and quaternary structure changes in allosteric proteins and interpreted their results in terms of a “global communication network” (GCN). These authors, however, defined allosteric any protein presenting heterotropic regulation: non-competitive enzyme inhibition, sequential and allosteric cooperativity, ligand-induced association-dissociation, all fit their definition of allostery which is even larger than that originally proposed by Monod in 1963 [2], let alone the identification of allosteric with concerted proposed by Monod in 1965 [3]. They find that the GCN occurs predominantly at the quaternary level for some of the proteins of their set, predominantly at the tertiary level for other proteins, but is mixed or interdependent (correlated quaternary and tertiary) for the majority of the proteins. Surprisingly, they exclude hemoglobin, and do not find a clear result for ATCase, the two best demonstrated examples of truly allosteric proteins obeying Equation (2).

Johnson and Barford [39] carried out a comparative analysis of structural features of allosteric proteins. These authors did not distinguish sequential and concerted reaction schemes, but the proteins they analyzed are all known or putative cases of allosteric, concerted homotropic cooperativity (Hb, ATC, GP, and PFK). They rightly emphasize that the proteins considered are symmetric oligomers. Since their analysis lacks examples of proteins obeying a sequential reaction mechanism, these authors cannot discuss whether the reaction mechanism has identifiable structural determinants.

## Figures and Tables

**Figure 1 biomolecules-12-01651-f001:**
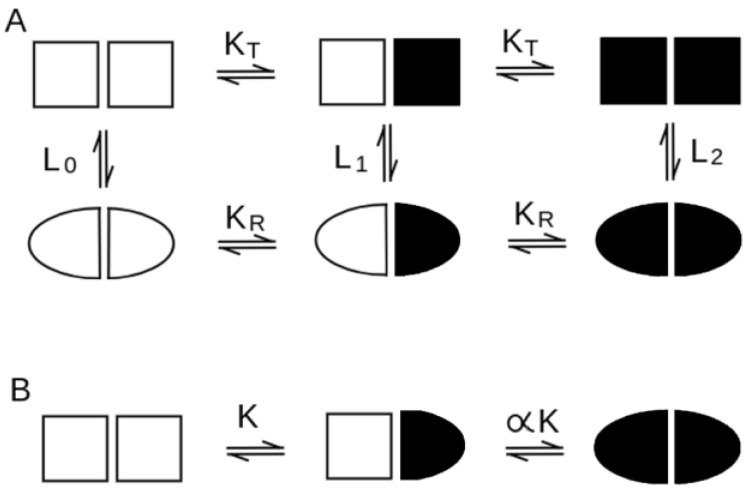
Schematic representation of the allosteric (**A**) and sequential (**B**) models of cooperativity. In both cases a cooperative homodimer is represented, in which full symbols represent liganded subunits, open symbols unliganded ones.

**Figure 2 biomolecules-12-01651-f002:**
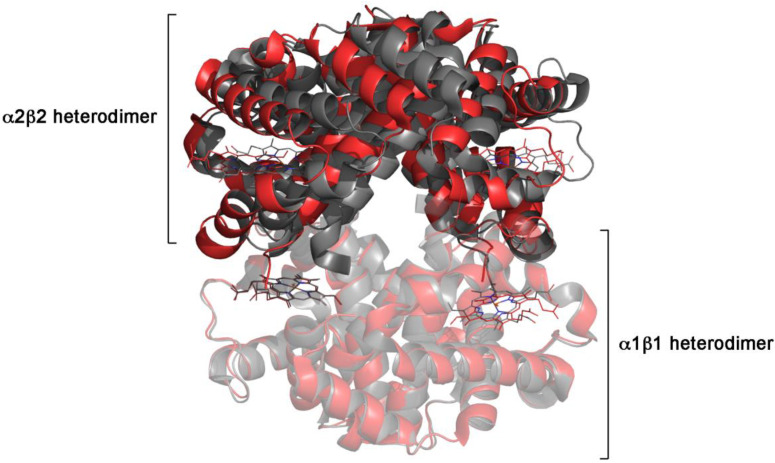
**Quaternary structure change in Hb.** Superposition of the α1β1 dimers from *R* state oxy Hb (PDB code 2DN1; pale red) and *T* state deoxy Hb (PDB code 2DN2; pale gray) shows that the tertiary structure changes are small; the large quaternary structure change is evident from the poor superposition of the α2β2 dimers from the same structures (vivid red and vivid gray).

**Figure 3 biomolecules-12-01651-f003:**
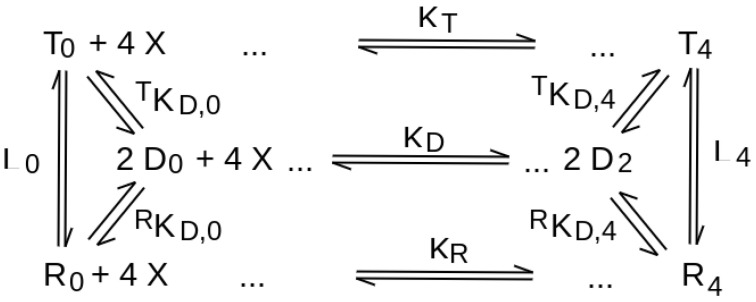
Relationships between ligand binding and dissociation into dimers for an allosteric tetramer.

**Figure 4 biomolecules-12-01651-f004:**
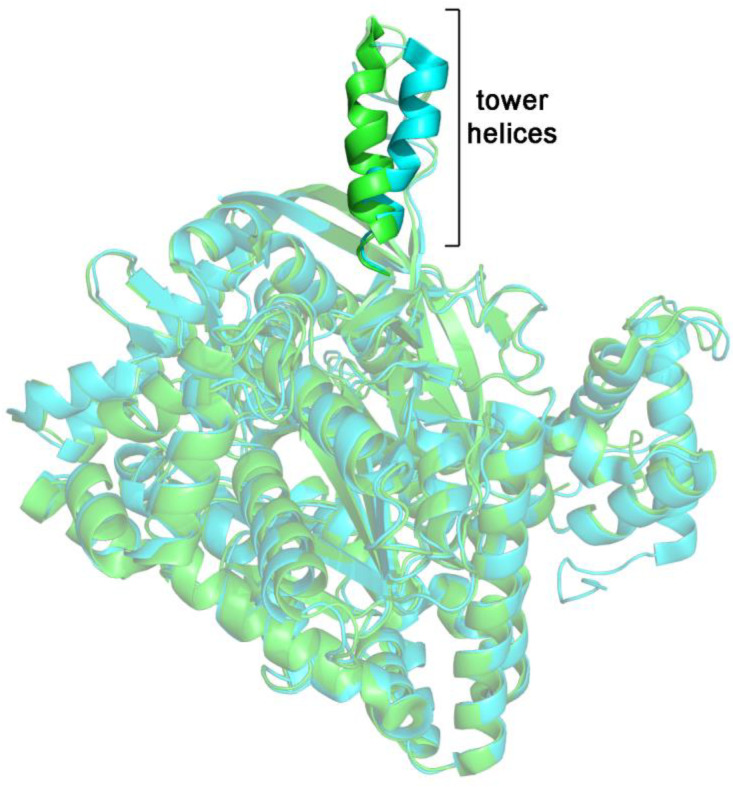
Superposition of monomers from *R* state (brown, pdb code 7GPB) and *T* state rabbit glycogen phosphorylase B (blue, pdb code 8GPB). Notice the change in the orientation of the tower helix, the uppermost structure in the figure (vivid colors).

**Figure 5 biomolecules-12-01651-f005:**
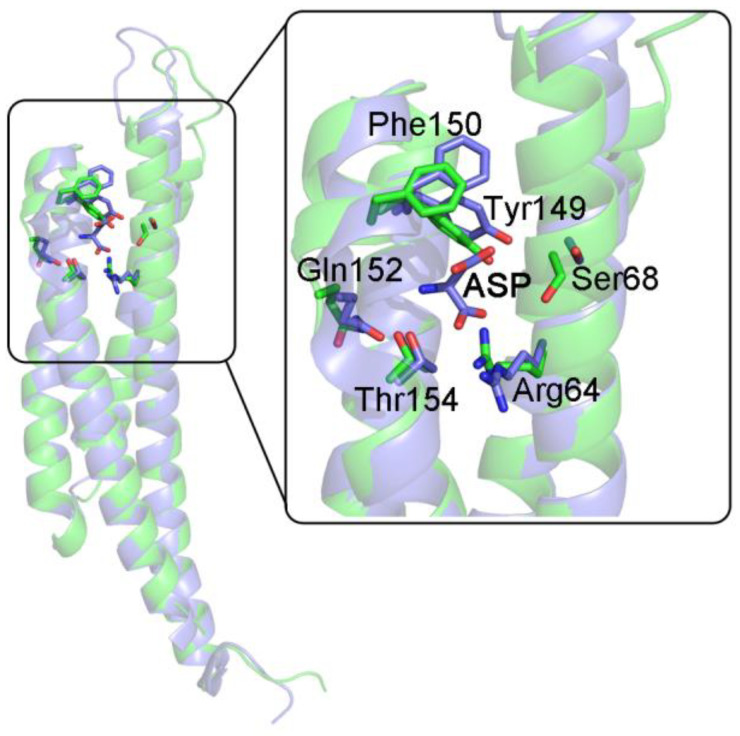
Superposition of the unliganded and liganded subunits of the Asp receptor from *S. typhimurium*. The two subunits from the asymmetric half-liganded homodimer of the Asp receptor (pdb code 1LIH) are rotated and superimposed onto each other. The position of the residues in the Asp binding site in the unliganded subunit (green) causes the site to shrink with respect to the liganded subunit (blue), where the site is occupied by Asp.

**Figure 6 biomolecules-12-01651-f006:**
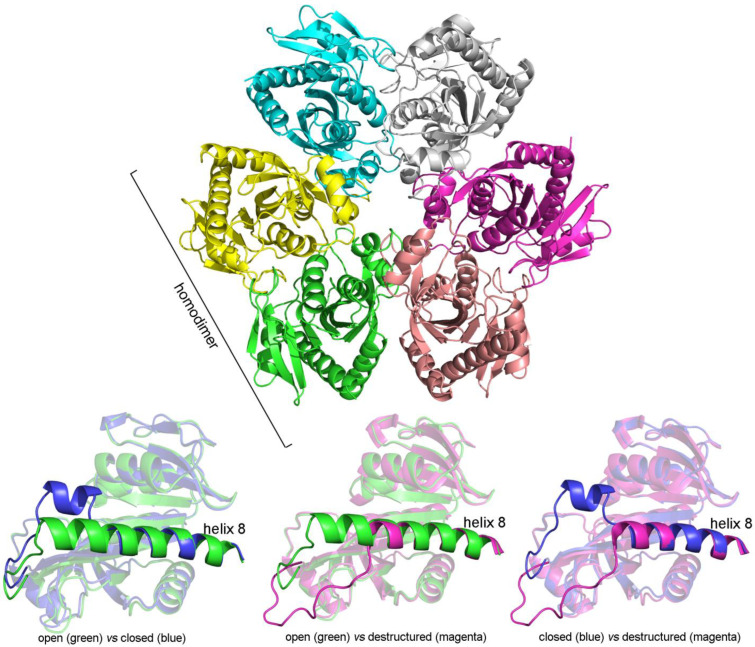
Quaternary structure of *E. coli* PNPase (upper panel) and tertiary structures of its subunits in different states of ligation (lower panels).

**Table 1 biomolecules-12-01651-t001:** Similarities and differences in the models of cooperativity.

	Concerted Models	Sequential Models
tertiary structure symmetry (all subunits share or do not share the same tertiary structure in the various ligation states)	necessary for unliganded, partially liganded and fully liganded states (asymmetry within *T* may be tolerated)	necessary asymmetry of ligation intermediates; asymmetry in the liganded state is tolerated

quaternary conformation equilibrium in the absence of the ligand	necessary, for all ligation intermediates, described by the allosteric constant L (see Equation (2)).	absent (see Equation (1))

cooperativity in the absence of quaternary structure change	impossible	possible

positive homotropic cooperativity	possible	possible

negative homotropic cooperativity	usually impossible (some exceptions may be considered)	possible

heterotropic regulation	almost always present	possible but not common

quaternary enhancement/constraint	usually quaternary constraint (but enhancement possible)	both possible

structural differences between the fully liganded and fully unliganded states	present (the main difference occurs between the *R* and *T* state, irrespective of ligation)	present

**Table 2 biomolecules-12-01651-t002:** List of proteins analyzed in this work.

Protein	Reaction Mechanism	Available Information
vertebrate hemoglobins	concerted	-positive homotropic cooperativity -ligand-independent allosteric equilibrium -symmetry of ligation intermediates -absence of cooperativity under conditions that prevent the quaternary structure change -quaternary constraint

Asp transcarbamylase	concerted	-positive homotropic cooperativity -ligand-independent allosteric equilibrium -quaternary constraint

Phosphofructokinase	likely concerted	-positive homotropic cooperativity -quaternary enhancement -*T* state is only induced by heterotropic effector (s) -Ligand binds only to the *R* state

Glycogen phosphorylase B	likely concerted	-positive homotropic cooperativity

Bacterial Asp receptor	sequential	-negative homotropic cooperativity -asymmetry of ligation intermediates

dTMP synthase	sequential	-negative homotropic cooperativity -asymmetry of ligation intermediates

Asp semialdehyde dehydrogenase	sequential	-negative homotropic cooperativity -asymmetry of ligation intermediates

Bacterial D-Lactate dehydrogenases	sequential	-negative or positive homotropic cooperativity -strong asymmetry in either the liganded or unliganded state

Bacterial Purine nucleoside phosphorylase	sequential	-negative homotropic cooperativity -strong asymmetry in the liganded state

**Table 3 biomolecules-12-01651-t003:** Structural changes associated with either ligand binding or intramolecular asymmetry are measured by Root Mean Square Deviation (RMSD) values calculated after optimal structure superposition of equivalent Cα atoms.

Ligation State RMSD	
Human Hemoglobin (Concerted, MWC-like)	
^R^ Hb(CO)_4_, PDB 2DN3, vs. ^T^ Hb, PDB 2DN2	tertiary, average: 0.67 Å overall (tertiary + quaternary): 2.43 Å
α1 subunit in ^T^ Hb, PDB 2DN2, vs. α2 subunit in ^T^Hb, PDB 2DN2	tertiary: 0.23 Å
β1 subunit in ^T^ Hb, PDB 2DN2, vs. β2 subunit in ^T^ Hb, PDB 2DN2	tertiary: 0.3 Å
^R^ Hb(CO)_4_, PDB 2DN3, vs. ^T^ α^FeCO^_2_ β^Co^_2_, PDB 1COH	tertiary, average: 0.67 Å
^R^ Hb(O_2_)_4_, PDB 2DN1, vs. ^T^Hb(O_2_)_4_, PDB 1GZX	overall (tertiary + quaternary): 2.57 Å
^T^ Hb, PDB 2DN2, vs. ^T^ Hb(O_2_)_4_, PDB 1GZX	tertiary, α subunits: 0.28 Å tertiary, β subunits: 0.41 Å overall (tertiary + quaternary): 0.51 Å
*E. coli* Asp transcarbamylase (concerted, MWC-like)	
Unliganded ^T^ ATC, PDB 6AT1 vs. PALA-liganded ^R^ ATC, PDB 8ATC	tertiary (catalytic subunits): 1.85 Å tertiary (regulatory subunits): 1.70 Å overall (tertiary + quaternary): 6.20 Å
Unliganded ^T^ ATC, PDB 6AT1 vs. unliganded ^T^ ATC, PDB 6AT1	tertiary (catalytic subunits): 0.56 Å tertiary (regulatory subunits): 1.34 Å
PALA-liganded ^R^ ATC, PDB 8ATC vs. PALA-liganded ^R^ ATC, PDB 8ATC	tertiary (catalytic subunits): 0.32 Å tertiary (regulatory subunits): 1.43 Å
*Geobacillus stearothermophylus* phosphofructokinase (concerted, MWC-like ?)	
PGC-inhibited ^T^ PFK, PDB 6PFK, vs. unliganded ^R^ PFK, PDB 3PFK	tertiary, average: 0.94 Å overall (tertiary + quaternary): 1.55 Å
PGC-inhibited ^T^ PFK, PDB 6PFK, vs. PGC-inhibited ^T^ PFK, PDB 6PFK	tertiary, average: 0.28 Å
Rabbit (*O. cuniculus*) muscle glycogen phosphorylase b (concerted, MWC-like ?)	
^R^ GPb, PDB 7GPB vs. ^T^ GPb, PDB 8GPB	tertiary, average: 1.33 Å overall (tertiary + quaternary): 2.61 Å
^R^ GPb, PDB 7GPB vs. ^R^ GPb, PDB 7GPB	tertiary, average: 0.55 Å
*S. typhimurium* Asp receptor (sequential, KNF-like)	
unliganded, PDB 1LIH vs. liganded, PDB 2LIG	tertiary 1LIH vs. 2LIG A: 2.58 Å tertiary 1LIH vs. 2LIG B: 2.42 Å overall (tertiary + quaternary): 2.85 Å
liganded, PDB 2LIG A (high affinity), vs. liganded, PDB 2LIG B (low affinity)	tertiary: 2.44 Å
*E. coli* Asp semialdehyde dehydrogenase (sequential, KNF-like)	
unliganded, PDB 1GL3, vs. half-liganded, PDB 1T4B	tertiary, liganded subunit of 1T4B: 1.3 Å tertiary, unliganded subunit of 1T4B: 0.94 Å overall (tertiary + quaternary): 1.49 Å
unliganded, PDB 1GL3, vs. unliganded, PDB 1GL3	tertiary: 0.27 Å
half-liganded, PDB 1T4B, vs. half-liganded, PDB 1T4B	tertiary: 0.62 Å
*P. aeruginosa* D-lactate dehydrogenase (sequential, KNF-like)	
unliganded PDB 6ABJ, vs. liganded PDB 5Z20	tertiary (average): 2.35 Å overall (tertiary + quaternary): 3.30 Å
unliganded PDB 6ABJ, vs. unliganded PDB 6ABJ	tertiary (average): 0.31 Å
liganded PDB 5Z20, vs. liganded PDB 5Z20	tertiary (average): 0.50 Å
*F. nucleatum* D-lactate dehydrogenase (sequential, KNF-like)	
unliganded PDB 6ABI, vs. liganded PDB 5Z21	tertiary (average): 4.7 Å overall (tertiary + quaternary): 6.44 Å
unliganded PDB 6ABI, vs. unliganded PDB 6ABI	tertiary (average): 1.9 Å
liganded PDB 5Z21, vs. liganded PDB 5Z21	tertiary (average): 0.25 Å
*E. coli* purine nucleoside phosphorylase (sequential, KNF-like)	
unliganded, PDB 1ECP, vs. fully liganded, PDB 1A69	tertiary (average) 0.88 Å tertiary (open vs. open) 0.31 Å tertiary (open vs. closed) 1.0–1.32 Å overall (tertiary + quaternary) 0.84 Å
unliganded, PDB 1ECP, vs. partially liganded, PDB 4TTA	tertiary (average) 1.3 Å tertiary (open vs. open) 0.47 Å tertiary (open vs. closed) 1.37 Å tertiary (open vs. destructured) 2.75 Å overall (tertiary + quaternary) 1.50 Å
fully liganded, PDB 1A69, vs. partially liganded, PDB 4TTA	tertiary (average) 1.44 Å tertiary (open vs. open) 0.45 Å tertiary (open vs. closed) 1.5 Å tertiary (closed vs. destructured) 3.5 Å tertiary (open vs. destructured) 2.75 Å overall (tertiary + quaternary) 1.71 Å
unliganded, PDB 1ECP, vs. unliganded, PDB 1ECP	tertiary (average) 0.32 Å
fully liganded, PDB 1A69, vs. fully liganded, PDB 1A69	tertiary (average) 1.35 Å tertiary (A-closed vs. B-open) 1.32 Å tertiary (A-closed vs. C-open) 1.76 Å tertiary (B-open vs. C-open) 0.98 Å
partially liganded, PDB 4TTA, vs. partially liganded, PDB 4TTA	tertiary (A-closed vs. F-closed) 0.17 Å tertiary (B-open vs. D-open) 0.45 Å tertiary (A/F-closed vs. B/D-open) 1.39 Å tertiary (A/F-closed vs. E-destruct.) 3.52 Å tertiary (B/D-open vs. E-destruct.) 2.82 Å

**Table 4 biomolecules-12-01651-t004:** Relative importance of tertiary structure changes with respect to overall ligand-induced structure changes. Structure changes are expressed as average RMSDs. Tertiary changes are those occurring between single subunits. Overall structure changes comprise the sum of tertiary and quaternary changes induced by ligation in the oligomers. R1: relative importance of the ligand-induced tertiary structure changes, measured as the averaged tertiary RMSDs in the liganded vs. unliganded state divided by the overall RMSD in the liganded vs. unliganded state. R2: asymmetry of subunits in the unliganded state, measured as the averaged tertiary RMSDs in the unliganded state divided by the overall RMSD in the liganded vs. unliganded state. R3: asymmetry of subunits in the liganded state, measured as the averaged tertiary RMSDs in the liganded state divided by the overall RMSD for liganded vs. unliganded state. *: measured for the catalytic subunits only.

Protein	Reaction Scheme	R1	R2	R3
human HbA (2DN2/2DN3)	concerted	0.28	0.11	0
*E. coli* ATC (6AT1/8ATC)	concerted	0.28	0.09 *	0.05 *
*G. stearothermophyilus* PFK (6PFK/3PFK)	likely concerted	0.60	0.18	0
rabbit glycogen phosphorylase b (7GPB/8GPB)	likely concerted	0.51	0	0.21
*S. typhimurium* Asp receptor (1VLT/1VLS)	sequential	0.88	0	0.85
*E. coli* Asp semialdehyde dehydrogenase (1T4B/1GL3)	sequential	0.75	0.18	0.42
*Pseudomonas aeruginosa* D-lactate dehydrogenase (6ABJ/5Z20)	sequential	0.71	0.09	0.15
*Fusobacterium nucleatum* D-lactate dehydrogenase (6ABI/5Z21)	sequential	0.73	0.29	0.04
*E. coli* purine nucleotide phosphorylase (1ECP/1A69)	sequential	1.05	0.38	1.60

## Data Availability

The coordinates of all the structures analyzed in this work are publicly available from the RCSB Protein Data Bank web site (https://www.rcsb.org/, accessed on 20 September 2022).

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
