# Peer review of "Structural Basis of Sequential and Concerted Cooperativity"

_biomolecules, 2022, doi:10.3390/biom12111651_

Round 1

Reviewer 1 Report

Dear Authors,

I really like the way manuscript is compiled with relevant enzyme systems. Introduction was elaborative and clear. Results section was explained in extensive and clear manner. However, I would suggest the authors to prepare good quality images with appropriate annotations to depict structural changes in unliganded and liganded states. I presume that a good quality image would convey most of the information to reader in a quick and efficient manner. I also felt like the data discussed in the manuscript is focused primarily on static state of structural conformations deposited in the PDB database. However, several dynamic structural transitions happen under physiological conditions in our body. So, I would recommend the authors to put forth a paragraph on these aspects and try to summarize few notable MD simulation studies demonstrating the allosteric phenomena. Finally, I really like the manuscript and data presented.

Author Response

We thank the reviewer for his/her comments.

We added a figure of the Asp receptor, to highlight the structure changes in the ligand binding site.

The reviewer is right in pointing out that relevant information on some of the systems we studied was provided by MD simulations, and, we may add, by rapid kinetic experiments; however, we are not experts in the field of MD simulations, this type of information is only available for a minority of the proteins and enzymes we considered, and is not sufficient for the comparative analysis that makes to core of our contribution. We added some information where pertinent and a short summary in the discussion. Essentially, the information that is most relevant to our point (i.e. the comparison between proteins obeying the concerted or sequential reaction mechanism) is that quaternary structure changes usually occur on a 100- to 1000- fold longer time window than tertiary structure changes. This has been very clearly demonstrated for Hb (with experimental data by Q.H. Gibson, W.A. Eaton and many others, and simulations by M. Karplus and co-workers, and again many other authors). Thus, the structural relaxations associated to ligand binding might be indicative of the reaction mechanism, either concerted (i.e. mainly quaternary) or sequential (i.e. mainly tertiary). Unfortunately, aside from this very basic finding, essentially every cooperative system has its own structural relaxation paths, and deriving general rules to be correlated with the reaction mechanism seems premature. 

Reviewer 2 Report

I provide my review report for the manuscript by Veronica Morea et al. titled "Structural basis of sequential and concerted cooperativity". This study is purely about the analysis of available structural data for a set of multi-domain proteins, and the characterization of structure changes in terms of RMSD, to distinguish and group allosteric from sequential action, as classified by traditional MWC and KNF models.
I have to admit that since my own work on protein allostery is focused on molecular dynamics, communication pathways, causality and information theory, I was not blown away by the findings presented by the authors. Nonetheless, I think that the analysis of available structural data is important and the authors' attempt to provide a more general classification as opposed to the majority of works digging into the details for a specific protein is worth to be shared with the allostery community. I therefore support publication.

I want the authors to check the equation on page 10, after line 327:
"For the above system, the law of mass conservation dictates:"
There is a problem with the dimensions in the first "[] bracket". The
first term in that bracket (1+x/K_R)^4 and the second term in that
bracket L_0(1+x/K_T)^4 do not have the same dimension (because of L_0),
but they should have the same dimension.

Author Response

As the reviewer correctly points out our main focus is on a comparative analysis on the relationship between reaction mechanism and structure changes of cooperative proteins. This is the reason why a model-free, general approach was selected: indeed RMSDs can be easily measured for every protein whose 3D structures are available, whereas more refined analyses tend to yield results that may be difficult to compare across different systems. As explained in the answer to reviewer 1, we added some pertinent information on MD simulations for some of the systems we analyzed.

L_0 in the original definition of the MWC model is dimension-less, being the equilibrium constant of the isomerization reaction that converts the unliganded R state protein to the T unliganded state protein. 

Author Response

We thank the reviewer for hs/her extended and detailed comments.

1.The term "allostery" has been used in the literature with different meanings, and we agree with the reviewer that "concerted" and "sequential" convey much more precise meanings that find unequivocal mathematical expressions in equations 1 and 2. Thus, whenever appropriate, we replaced allosteric with concerted, notably in all entries of Tables 1 and 2. However, we remark that while Monod explicitly linked the concept of allostery to his concerted model of 1965 (eq. 2; the title of the paper is "On the nature of allosteric transitions: a plausible model" and the authors state: "Two (at least two) states are reversibly accessible to allosteric oligomers"), Koshland never used the term allostery to refer to the sequential model he formalized in 1966 (eq. 1). Thus, different authors used the term allostery with different meanings.

2. We moved Fig. 1 in order to be quoted in the text before equations 1 and 2. The meaning of the terms in the equations is now better described, and P(x) is defined. Association constants were used by Adair, Pauling, and Koshland (among others), whereas Monod used dissociation constants; thus either formalism should be modified to achieve consistency of equations 1 and 2. We accepted the reviewer’s suggestion, and converted dissociation into association constants.

3 and 4. Table 1 is of crucial relevance to our work, because the first problem we had to solve was: which cooperative proteins are true and proven examples of either the concerted or sequential mechanism? Table 1 lists the definition criteria of the two mechanisms, which however may be difficult to access experimentally. We clarified some definitions, but essentially we cannot change it very much; however, we explain the function and relevance of the Table in greater detail in the text of the revised manuscript. 

Table 2 contains the assignment of the reaction mechanism to the proteins we selected. In the case of Hb and ATCase the quaternary structure change in the absence of changes in the ligation state has been proven with certainty; moreover in Hb the structure of diliganded intermediates is known, is T state and symmetric. Thus Hb and ATCase obey a concerted reaction mechanism. PFK and GP are reported as "probably concerted" in Table 2 because of the large quaternary structure change and literature consensus, but the evidence is not as strong as for Hb and ATCase. 

The sequential reaction mechanism may explain both positive and negative homotropic cooperativity, but the former is compatible also with a concerted reaction mechanism, and the attribution of a concerted or sequential reaction mechanism to positively cooperative systems is often uncertain. By contrast, negative cooperativity is not compatible with the concerted mechanism (except in the presence of chemical inequality of the subunits: e.g. Root effect Hbs, or structural asymmetry of the R state), thus we selected negatively cooperative proteins as proven or at least likely cases of the sequential reaction mechanism. In this decision we follow the most conservative and prudent strategy. In no way do we imply that a sequential reaction mechanism cannot cause positive cooperativity, and indeed one enzyme in our set is considered sequential and has positive cooperativity, namely FnLDH, whose similarity with the negatively cooperative PaLDH justifies our attribution of the mechanism. This has been better explained in the text. 

5 and 6 Tables 3 and 4 and their legends have been improved, the apparent lack of the PDB code was due to a formatting error occurring at the end of the page; it should disappear in the printed version of the paper. The meaning of ratios R1, R2 and R3 is explained in the table legend; unfortunately it is too long to appear in the column heading.

7. The legend of Fig. 2 has been improved. The reviewer is right: the alpha1beta1 dimer of THb and RHbO2 has been optimally superimposed, and the alpha2beta2 of the two structures show the rotation and shift associated to the quaternary structure difference.

8 and 9. A figure for Asp receptor has been added

10. The discussion of quaternary enhancement on pages 9 and 10 is important and we explained it better in the introduction. It was confined to the legend of Fig. 2 in order not to overload the text, but we think that it should stay. The reason is given above (see the discussion of points 3 and 4). We are aware of the work by S. Edelstein, G.K. Ackers and others, but we are also aware that these authors substantially disagree on this point, and G. Weber disagrees with both of them.

Monod et al. (1965), Gibson and Edelstein (1987) and Changeux (2013) stated that the concerted model demands quaternary constraint, and is incompatible with quaternary enhancement ("The conformation of each protomer is constrained by its association with the other protomers", MWC 1965; "This effect, which they [Ackers et al.] called quaternary enhancement, is incompatible with the two-state Monod, Wyman, and Changeux allosteric model", Gibson and Edelstein 1987; "... what I considered the fundamental mechanistic issue of the model: the cooperative symmetrical assembly of the subunits within an allosteric oligomer and the quaternary constraint established between subunits ...", Changeux 2013). If we followed this view we would have been allowed to state with certainty that cooperative proteins presenting quaternary enhancement obey a sequential mechanism, and we would not have limited our examples of sequential proteins to negatively cooperative ones. Since, as a general rule proteins presenting the ligand binding site at an intersubunit interface have quaternary enhancement, this would have simplified our work considerably. Some cases would have been sensational: PFK would have been assigned a sequential, rather than concerted, reaction mechanism. However, as we demonstrate in the legend of Fig. 3, this assumption is wrong: the concerted model is compatible with quaternary enhancement. Ackers was aware of this fact, but he only discussed it in relation to Hb ("Gibson and Edelstein have advanced an argument ... that the existence of quaternary enhancement would disprove the two-state MWC model of allosteric regulation. In fact, Ackers and Johnson ... had explicitly demonstrated the exact opposite: the subunit assembly-linked oxygenation data ... was found to provide an excellent fit to a model in which the tetramers conform strictly to the two-state MWC model while the oxygen affinity of dissociated dimers is lower than that of R-state tetramers" Ackers and Johnson 1990).

In conclusion, in the legend of Fig. 3 we solve a long standing issue, provide a general mathematical demonstration that quaternary enhancement is compatible with a concerted reaction mechanism, and justify the otherwise puzzling attribution of this mechanism to PFK and possibly other proteins having the ligand binding site at an intersubunit interface. We think that this is an important added value of our manuscript, and that removing it would cause doubts in all readers who strictly follow Monod's postulates.

11. The work of Daily and Gray is relevant and duly quoted, but does not provide the information we look for, because the authors analyze "allosteric proteins” without any attempt to discriminate their reaction mechanism. Actually their usage of the word "allosteric" is not clearly defined and they seem to understand it in the sense of Monod et al. [1963], ignoring all subsequent work. Consistently with their premises, Daily and Gray are uninterested in the distinction between sequential and concerted mechanisms, which is at the center of our analysis. Indeed we think that for the majority of the proteins they analyzed the reaction mechanism cannot be attributed with reasonable certainty. Moreover the two best characterized and most certain instances of concerted proteins, Hb and ATCase, are poorly treated in their paper: "While hemoglobin exhibits homotropic connectivity among the four hemes ... it is excluded because it is not heterotropic" (and we disagree with this statement, even if the definition of allostery by Daily and Gray is adopted, because Hb has heterotropic effectors, notably but not only DPG); ATCase in their analysis has 0 tertiary network, 0 quaternary network, 0 global network and its communication class is undefined (see Table 1 of Daily and Gray 2009). Thus, their results and ours do not overlap enough to warrant a more detailed comparison.